

# Inter-residue through-space scalar $^{19}$F–$^{19}$F couplings between CH$_2$F groups in a protein

Yi Jiun Tan[1], Elwy H. Abdelkader[1], Iresha D. Herath[1], Ansis Maleckis[2], Gottfried Otting[1]

[1] ARC Centre of Excellence for Innovations in Peptide & Protein Science, Research School of Chemistry, Australian National University, Canberra, ACT 2601, Australia

[2] Latvian Institute of Organic Synthesis, Aizkraukles 21, LV-1006 Riga, Latvia

*Correspondence to*: Gottfried Otting (gottfried.otting@anu.edu.au)

**Abstract.** Using cell-free protein synthesis, the protein G B1-domain (GB1) was prepared with uniform high-level substitution of leucine by (2$S$,4$S$)-5-fluoroleucine, (2$S$,4$R$)-5-fluoroleucine, or 5,5'-difluoroleucine. $^{19}$F nuclear magnetic resonance (NMR) spectra showed chemical shift ranges spanning more than 9 ppm. Through-space scalar $^{19}$F–$^{19}$F couplings between CH$_2$F groups arising from transient fluorine–fluorine contacts are readily manifested in [$^{19}$F,$^{19}$F]-TOCSY spectra. The $^{19}$F chemical shifts correlate with the three-bond $^1$H–$^{19}$F couplings ($^3J_{HF}$), confirming the γ-gauche effect as the predominant determinant of the $^{19}$F chemical shifts of the CH$_2$F groups. Different $^3J_{HF}$ couplings of different CH$_2$F groups indicate that the rotation of the CH$_2$F groups can be sufficiently restricted in different protein environments to result in the preferential population of a single rotamer. The $^3J_{HF}$ couplings also show that CH$_2$F groups populate the different rotameric states differently in the 5,5'-difluoroleucine residues than in the monofluoroleucine analogues, showing that two CH$_2$F groups in close proximity influence each other's conformation. Nonetheless, the $^{19}$F resonances of the C$^{δ1}$H$_2$F and C$^{δ2}$H$_2$F groups of difluoroleucine residues can be assigned stereospecifically with good confidence by comparison with the $^{19}$F chemical shifts of the enantiomerically pure fluoroleucines. $^1$H–$^{19}$F NOEs observed with water indicate hydration with subnanosecond residence times.

## 1 Introduction

Proteins made with global substitution of a single amino acid type by a selectively fluorinated analogue greatly facilitate their analysis by $^{19}$F-NMR spectroscopy (Sharaf and Gronenborn, 2015). Structural perturbations caused by the fluorine substitutions can be kept to a minimum if a single fluorine atom is installed in a methyl group, as the resulting CH$_2$F group has the freedom to respond to the increased spatial requirement of the C–F moiety by preferential population of those rotamers that are most readily accommodated by the chemical environment. Recently, we showed that the *E. coli* peptidyl-prolyl isomerase B (PpiB), which contains five leucine residues, can be produced with high-level uniform substitution of leucine for (2$S$,4$S$)-5-fluoroleucine (FLeu1), (2$S$,4$R$)-5-fluoroleucine (FLeu2) or 5,5'-difluoroleucine (diFLeu; Fig. 1) by using cell-free protein synthesis (Tan et al., 2024). As demonstrated by X-ray crystal structures, the structural perturbations caused by these amino acid substitutions were minimal (Frkic et al., 2024a). Furthermore, the $^3J_{HF}$ coupling constants were inversely correlated with the $^{19}$F chemical shifts, in a first experimental confirmation of the γ-gauche effect predicted by Oldfield and co-workers



based on quantum calculations (Feeney et al., 1996). In the structure of PpiB, the leucine residues are isolated from each other. In contrast, the three leucine residues of GB1 are arranged such that methyl groups of neighbouring leucine residues can make van der Waals contacts (Fig. 2). This situation may produce through-space scalar $J_{FF}$ ($^{TS}J_{FF}$) couplings.




**Figure 1.** Chemical structures of the fluorinated leucine analogues used in the present work. (2S,4S)-5-fluoroleucine, (2S,4R)-
5-fluoroleucine and 5,5'-difluoro-L-leucine are referred to in the following as FLeu1, FLeu2 and diFLeu, respectively.

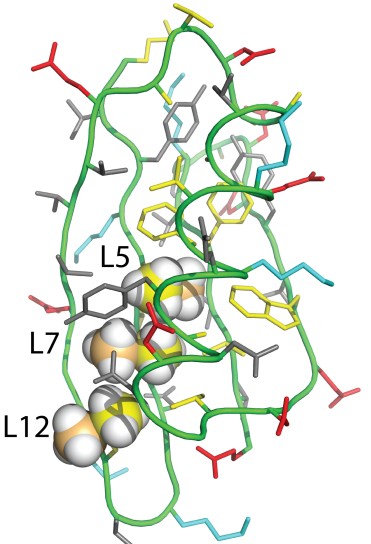

**Figure 2.** Solution structure of GB1 (PDB ID 3GB1, Juszewski et al., 1999). The methyl groups of Leu5, Leu7 and Leu12 are shown in a space filling representation with the $\delta_1$- and $\delta_2$-carbon atoms in orange and yellow, respectively. The side chain
of Leu5 is inaccessible to solvent, whereas the $C^{\delta 1}H_3$ groups of Leu7 and Leu12 are partially and highly accessible, respectively.

**MAGNETIC RESONANCE**
Open Access Discussions

## 2 Experimental procedures

### 2.1 Fluorinated leucine analogues

Initially, the fluorinated leucine analogues FLeu1 and FLeu2 (Fig. 1) were synthesized following published protocols (Moody
et al., 1994; August et al., 1996; Charrier et al., 2004). diFLeu with and without $^2$H substitutions was synthesized as described
(Maleckis et al., 2022). Subsequently, FLeu1, FLeu2 and diFLeu were obtained as HCl salts from Enamine (Ukraine).

### 2.2 Expression vectors

Expression vectors were based on pETMCSI (Neylon et al., 2000) and constructed with a C-terminal His$_6$ tag following a TEV
cleavage site. The amino terminus was preceded by the 5'-nucleotide sequence of the T7 gene 10 to ensure high expression
yields. The full nucleotide and amino acid sequences are shown in Table S1.

### 2.3 Protein expression

All protein samples were expressed by continuous exchange CFPS following an established protocol (Apponyi et al., 2008;
Ozawa et al., 2012). The gene of the GB1 construct was PCR-amplified with eight-nucleotide overhangs to generate
circularized DNA suitable for use in CFPS (Wu et al., 2007). Leucine was omitted when preparing the acid soluble amino acid
mixture. The fluoroleucine of interest was added from an aqueous stock solution to the outer buffer at a final concentration of
4 mM. The pH of the outer buffer was adjusted to 7.5. The CFPS reaction was conducted at 30 °C for 16 h using 1 mL inner
reaction mixture of S30 cell extract made from the *E. coli* BL21 strain and 10 mL outer buffer.

### 2.4 Protein purification

Proteins were purified using a 1 mL Ni–NTA gravity column (GE Healthcare, USA) equilibrated with buffer A (50 mM Tris–
HCl, pH 7.5, 300 mM NaCl), using buffer B (same as buffer A but with 10 mM imidazole) for washing and buffer C (same as
buffer A but with 300 mM imidazole) for elution. The purified proteins were dialyzed overnight against storage buffer (50
mM HEPES, pH 7.5, 100 mM NaCl) and concentrated using an Amicon centrifugal ultrafiltration tube with a molecular weight
cut-off of 3 kDa.

### 2.5 Protein mass spectrometry

Intact protein mass analysis was performed using an Orbitrap Elite Hybrid Ion Trap–Orbitrap mass spectrometer equipped
with an UltiMate 3000 UHPLC (Thermo Scientific, USA). The protein samples were injected via an Agilent ZORBAX SB-
C3 Rapid Resolution HT Threaded Column using a 5–80% gradient of acetonitrile with 0.1% formic acid. The data were

collected in positive ion mode. The protein masses were obtained by deconvolution using the Xtract function in the Qual Browser software tool of the program Xcalibur 3.0.63 (Thermo Fisher Scientific, USA).

### 2.6 Protein NMR conditions

All $^{19}$F-NMR spectra were measured at 25 °C on a 400 MHz Bruker Avance NMR spectrometer equipped with a SmartProbe, allowing $^{19}$F detection with $^{1}$H decoupling. The protein solutions were in 90% $H_2O$/10% $D_2O$ with 20 mM MES buffer, pH 6.5, and 100 mM NaCl or with 50 mM HEPES buffer, pH 7.5. 0.1 mM trifluoroacetate (TFA) was added as an internal reference for the samples and calibrated to -75.25 ppm.

## 3 Results

### 3.1 Protein yields and purity

Up to 2.7 mg of protein was obtained from 1 mL inner reaction mixture of the CFPS setup (Table S2). The amino acid sequence of native GB1 contains three leucine residues and the additional TEV cleavage site present in our constructs adds a fourth leucine residue. Intact protein mass spectrometry indicated that the predominant species contained fluorinated leucine

analogues at all four sites. The species containing three or two fluorinated leucine analogues were also detected, but their intensity indicated that the chance of canonical leucine at any of the four sites was below 10% (Fig. S1). Mass spectra of GB1 produced with diFLeu in the presence of some canonical leucine delivered the natural protein as the main species, followed by protein containing single leucine-for-diFLeu substitutions, illustrating the strong preference of the *E. coli* leucyl-tRNA synthetase for L-leucine over diFLeu (Fig. S2). Complete exclusion of L-leucine from the CFPS reaction could not be achieved

due to amino acid impurities in the S30 extract.

### 3.2 Protein stability

Thermal denaturation measured by circular dichroism at 216 nm showed that the melting temperatures the GB1 samples made with fluorinated leucine analogues ranged between about 66 °C and 72 °C, i.e. 9 – 15 degrees lower than for the wild-type protein (Fig. S3), indicating that the presence of $CH_2F$ groups decreases the stability of the protein.


### 3.3 1D $^{19}$F-NMR spectra

Figure 3 shows the 1D $^{19}$F-NMR spectra of the GB1 variants produced with diFLeu (GB1-d), FLeu1 (GB1-1) or FLeu2 (GB1-2). In addition, Fig. 3b shows the spectrum of GB1 produced with diFLeu in the presence of canonical L-leucine (GB1-dd). The 1D NMR spectra resolve the signals of all fluorine atoms.




The chemical shifts are very sensitive to the chemical environment. A striking illustration are the very different chemical shifts observed in GB1-d, when the sample was prepared with the addition of L-leucine to produce samples predominantly containing single diFLeu residues (GB1-dd; Fig. 3a and 3b). Comparison of the high-field and low-field ends of the spectra of GB1-dd and GB1-d shows that minor peaks observed for the GB1-d sample correspond to main peaks observed with GB1-dd and vice

versa. The minor peaks in Fig. 3a can thus be attributed to a small amount of canonical leucine in the protein preparation. Conversely, the minor peaks in the spectrum of GB1-dd appear to correspond to peaks of the fully fluorinated GB1-d sample, although the only minor species present in significant amounts contains no more than two diFLeu residues. This indicates that the presence of a second diFLeu residue is sensed only if it is in the immediate neighbourhood. Position 7 features two neighbouring leucine sites (Fig. 2), yet the $C^{\delta 1}H_2F$ group seems to sense predominantly a single neighbour, while the chemical

shift of the $C^{\delta 2}H_2F$ group is less well conserved between the major species in GB1-dd and the minor species in GB1-d (Fig. 3a and 3b).

In the case of the GB1-2 sample, minor peaks arose because the FLeu2 amino acid synthesized in-house contained about 10% FLeu1 as an impurity.

In the case of residue 59, which is in the flexible TEV protease recognition site of the C-terminal peptide segment of the protein

construct, the chemical shifts are hardly impacted by the rest of the protein as indicated by their conservation between the spectra of Fig. 3a and 3b. For the diFLeu residue in position 59 (Fig. 3a and 3b), we base the stereospecific assignment on the $^{19}F$ chemical shifts observed for this position in GB1-1 and GB1-2 (Fig. 3c and 3d). The stereospecific assignments of the other diFLeu residues were determined by 2D NMR experiments described below. They attributed the high-field signals of residues 5, 7 and 12 to the $C^{\delta 1}H_2F$ groups. Notably, the respective signals in GB1-1 are also high-field of the corresponding

signals in GB1-2 (Fig. 3c and 3d).

The $T_1$ relaxation times were of the order of 0.3 s and the full line widths at half-height ranged between 7 and 15 Hz. The broadest lines were observed for residue 5, the side chain of which is deeply buried in the core of the protein (Fig. 2), hence expected to feature the least flexibility and the fastest transverse relaxation rates. Residue 7 is the next most-buried residue while the side chain of residue 12 is more highly accessible to solvent (particularly the $C^{\delta 1}H_2F$ group, see Fig. 2) and residue

59 is completely solvent-exposed.

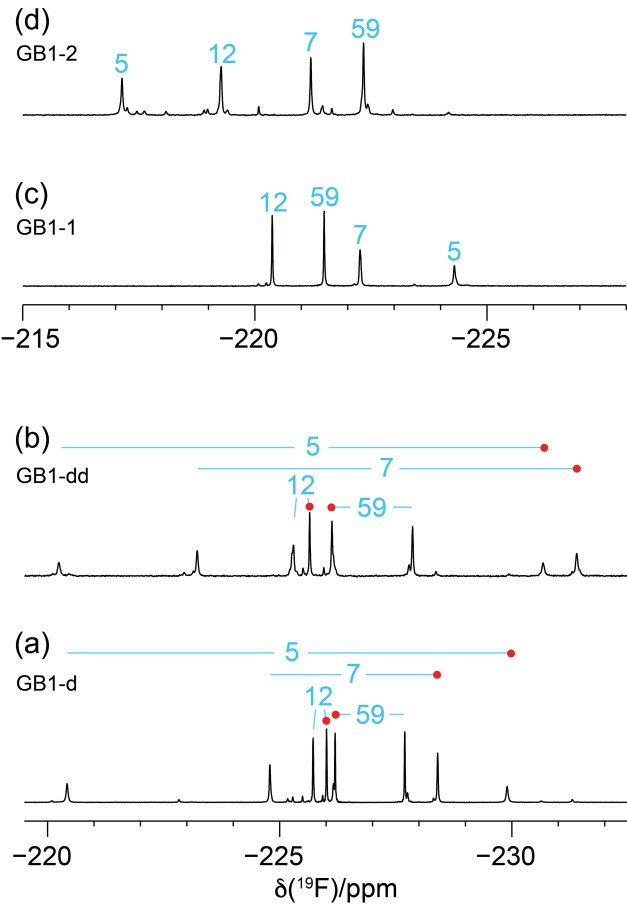

**Figure 3.** 1D $^{19}$F-NMR spectra of GB1 made with fluorinated leucine analogues, using FLeu1 to produce GB1-1, FLeu2 for GB1-2 and diFLeu for GB1-d and GB1-dd. All spectra were recorded with $^{1}$H decoupling during acquisition, using 0.5 s recovery delay between scans. The resonance assignments are indicated by the sequence numbers of the four leucine sites. (a) GB1-d prepared with 4 mM diFLeu. Spectrum recorded of a 2 mM protein solution in 20 mM MES buffer pH 6.5. Red dots mark the resonances assigned to $C^{\delta 1}H_2F$ groups. (b) GB1-dd prepared with a mixture of 0.5 mM leucine and 4 mM diFLeu. Spectrum recorded of a 4 mM protein solution in 20 mM MES buffer pH 6.5. Stereospecific assignments are indicated as in (a). (c) Spectrum of a 2 mM solution of GB1-1 in 50 mM HEPES pH 7.5. (d) Spectrum of a 2.2 mM solution of GB1-2 in 50 mM HEPES pH 7.5.

Recording of the $^{19}$F NMR spectra without decoupling of the $^{1}$H spins revealed broad multiplets with overlap between some of the resonances (Fig. 4). The multiplet of each CH$_2$F group is composed of a triplet of doublets due to 2-bond couplings, $^{2}J_{HF}$, within each CH$_2$F group (47 Hz) and the $^{3}J_{HF}$ coupling with the methine proton of the isopropyl group. $^{3}J_{HF}$ couplings obey a Karplus relationship (Williamson et al., 1968; Gopinathan and Narasimhan, 1971). If the $^{3}J_{HF}$ coupling is small, the





envelope of the multiplet appears like a triplet, but $^3J_{HF}$ can also be as large as 44 Hz (Tan et al., 2024), in which case the multiplet appears like a quartet. The $^{19}$F resonances of residue 5 in GB1-1 and GB1-2 are examples of these two limiting cases (Fig. 4c and 4d).

Interestingly, the multiplet of the $C^{\delta 1}H_2F$ group of residue 12 in GB1-dd displays narrower lines than the $C^{\delta 2}H_2F$ group (Fig. 4b), in agreement with a narrower signal in GB1-1 than in GB1-2 (Fig. 3). This observation aligns with the greater solvent exposure of the $C^{\delta 1}H_2F$ group (Fig. 2). The inverse correlation between $^{19}$F-NMR line width and solvent exposure suggests that faster rotation of the $CH_2F$ groups about the $C^\gamma$–$C^\delta$ bond results in slower transverse relaxation.

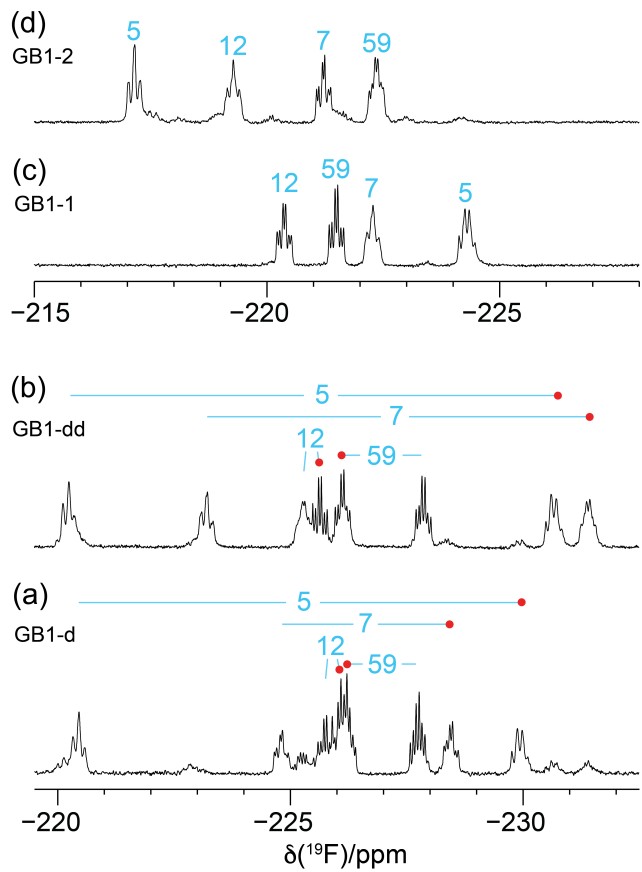

**Figure 4.** 1D $^{19}$F-NMR spectra recorded of (a) GB1-d made with diFLeu, (b) GB1-dd made with diFLeu diluted with canonical L-leucine, (c) GB1-1 made with FLeu1 and (d) GB1-2 made with FLeu2. The spectra were recorded without $^1$H decoupling. The resonance assignments are indicated in blue. Red dots identify stereospecific assignments of the $C^{\delta 1}H_2F$ groups in diFLeu residues.





### 3.4 NMR resonance assignments


The large $J_{HF}$ couplings observed indicate that resonance assignments can be achieved by coherence transfer between [1]H and [19]F spins and linking the [1]H resonances of the isopropyl groups to the backbone protons by [[1]H,[1]H]-TOCSY and [[1]H,[1]H]-NOESY spectra. The [1]H chemical shifts of the $CH_2F$ groups are near 4 ppm and the methine resonances are between 1 and 2 ppm. For GB1 made with diFLeu, a [[1]H,[19]F]-COSY spectrum connected the [19]F NMR signals belonging to the same residue

(Fig. 5).

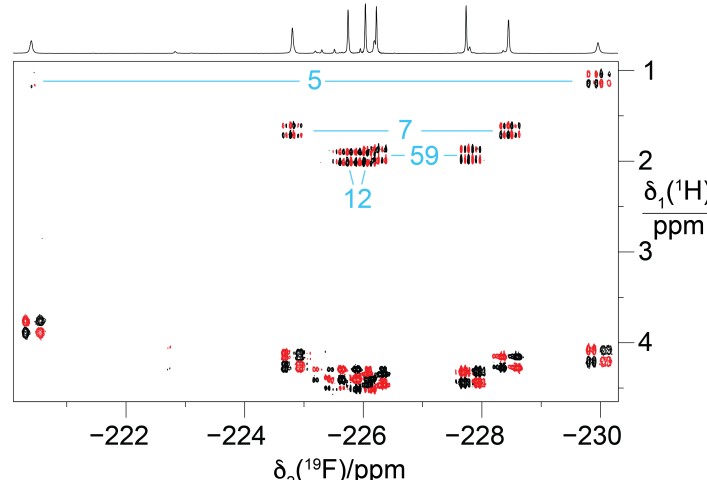

**Figure 5.** [[1]H,[19]F]-correlation spectrum of a concentrated solution of GB1-d (about 10 mM). The [1]H-decoupled 1D [19]F-NMR spectrum is shown at the top. The [[1]H,[19]F]-COSY spectrum was recorded with the pulse sequence 90°([1]H) – $t_1$ – 

90°([1]H),90°([19]F) – acquisition([19]F). The cross-peaks with the methine proton of the isopropyl groups, which identify the pairs of $CH_2F$ groups belonging to the same residue, are assigned in blue. Parameters: $t_{1max}$ = 51 ms, $t_{2max}$ = 217 ms, total recording time 1.4 h.

To probe for the presence of scalar through-space [19]F–[19]F couplings in GB1-d, we recorded a [[19]F,[19]F]-TOCSY spectrum. The

spectrum yielded both intra-residual and through-space correlations (Fig. 6a). Interestingly, the intra-residual cross-peak of residue 5 could not be observed, whereas the inter-residual connectivities with the nearest neighbour (residue 7) were intense. Residue 7 in turn showed cross-peaks to residues 5 and 12, which were more intense than the intra-residual cross-peaks. The absence of the intra-residual cross-peak of residue 5 indicates that scalar $^4J_{FF}$ couplings cannot be relied upon to connect the [19]F-NMR signals of the $CH_2F$ groups of each diFLeu residue.

Notably, some of the most intense [[19]F,[19]F]-TOCSY cross-peaks came about by $^{TS}J_{FF}$ couplings. To exclude the possibility of TOCSY cross-peaks arising from [19]F–[19]F NOEs, we also recorded a [[19]F,[19]F]-NOESY spectrum (Fig. 6b). The NOESY spectrum produced the intra-residual cross-peak of residue 5 with greater intensity than the inter-residual NOEs. This illustrates



the different dependence of NOEs and $^{TS}J_{FF}$ couplings on the internuclear distance, with $^{TS}J_{FF}$ couplings depending on close contacts between the fluorine atoms to create the necessary orbital overlap. Notably, although the NOESY spectrum had been

recorded of a GB1-d sample with over 10-fold higher protein concentration, the cross-peak intensities were markedly poorer in the NOESY than in the TOCSY spectrum.

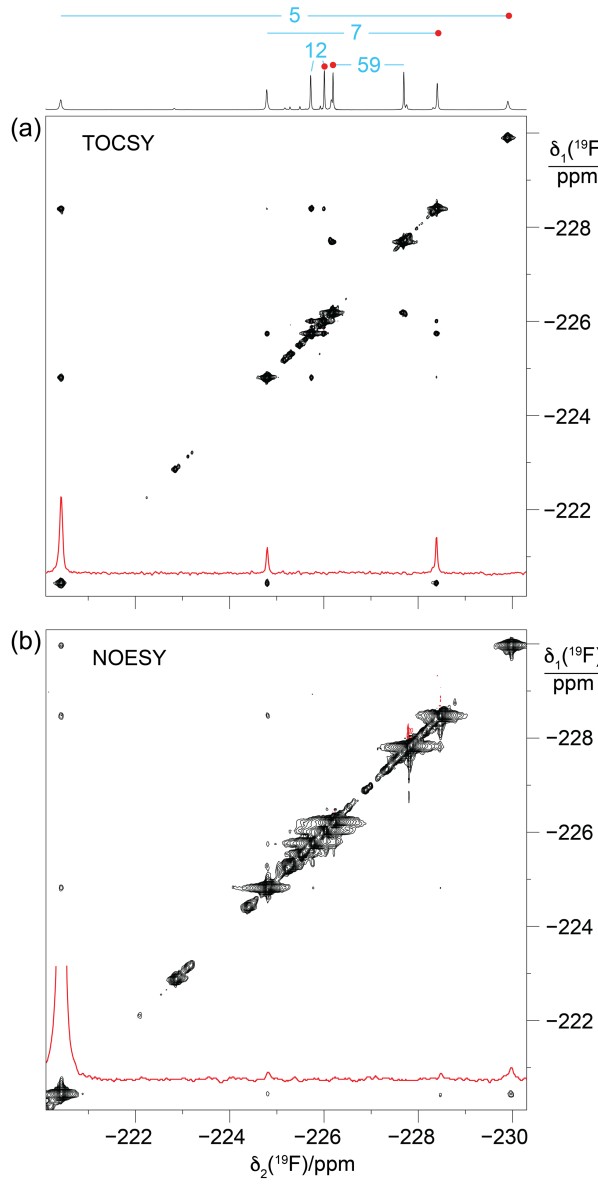

**Figure 6.** [$^{19}$F,$^{19}$F]-TOCSY and [$^{19}$F,$^{19}$F]-NOESY spectra of GB1-d. The 1D $^{19}$F-NMR spectrum is shown at the top along with the resonance assignments. Cross-sections through the diagonal peaks at -220.4 ppm are shown in red. (a) [$^{19}$F,$^{19}$F]-TOCSY

spectrum (mixing time 60 ms) recorded of an 0.8 mM protein solution. Parameters: $t_{1max} = 26$ ms, $t_{2max} = 105$ ms, total recording time 14 h. (b) [$^{19}$F,$^{19}$F]-NOESY spectrum (mixing time 200 ms) recorded of a > 10 mM solution of GB1-d. Parameters: $t_{1max}$





= 13.5 ms, $t_{2max}$ = 108 ms, total recording time 12 h, processed with 20 Hz exponential line broadening in the $d_2$ dimension. Without cropping the diagonal peak in the cross-section would exceed the boundaries of the figure.

The GB1 samples prepared with FLeu1 or FLeu2 (GB1-1 and GB1-2, respectively) offer fewer opportunities for $^{TS}J_{FF}$ couplings. The conformation shown in Fig. 2 excludes direct contacts between $C^{\delta1}H_3$ groups, whereas van der Waals contacts between $^{19}F$ atoms of $C^{\delta2}H_2F$ groups are arguably possible in view of the greater C–F bond length and larger van der Waals radius of fluorine compared with hydrogen. Even so, direct fluorine–fluorine contacts in GB1-2 depend on specific rotamer combinations of neighbouring $CH_2F$ groups and may be infrequent if the $CH_2F$ groups rotate.

Experimentally, residue 7 produced inter-residual cross-peaks in the $[^{19}F,^{19}F]$-TOCSY spectra of GB1-1 and GB1-2 (Fig. 7). In the case of GB1-1, the cross-peaks were about 100 times smaller than the diagonal peaks. In the case of GB1-2, residue 7 produced cross-peaks both with residue 5 and residue 12, and those cross-peaks were only about 10 times smaller than the diagonal peaks. No single conformation of the $C^{\delta2}H_2F$ group of residue 7 can simultaneously engage in fluorine–fluorine contacts with residues 5 and 12 (Fig. 2), suggesting that the $C^{\delta2}H_2F$ group populates different rotamers. Furthermore, the cross-
peak observed between the $C^{\delta1}H_2F$ groups of residues 7 and 12 in GB1-1 suggests that these side chains enjoy greater conformational freedom than suggested by the NMR structure 3GB1.

The fluorine–fluorine contacts observed in GB1-2 recapitulate the two strongest cross-peaks observed with the high-field $^{19}F$ resonance of residue 7 in GB1-d (Fig. 6a). Assuming that the side-chain conformations are conserved between GB1-2 and GB1-d, this affords stereospecific assignments of GB1-d, assigning the high-field signals of residues 7 and 12 and the low-
field signal of residue 5 to the $^{19}F$ spins of the respective $C^{\delta2}H_2F$ groups. Given this assignment, the weaker interaction between the low-field signals of residues 7 and 12 indicates a contact between a $C^{\delta1}H_2F$ and a $C^{\delta2}H_2F$ group, which cannot occur in either GB1-1 or GB1-2. The generally greater cross-peak intensities observed in GB1-d may be a consequence of the greater steric crowding associated with the spatial demands of multiple fluorine atoms, bringing the $^{19}F$ spins into closer contact. In addition, the greater density of $^{19}F$ spins in GB1-d opens the chance for multiple magnetisation transfer steps during the TOCSY
mixing period.



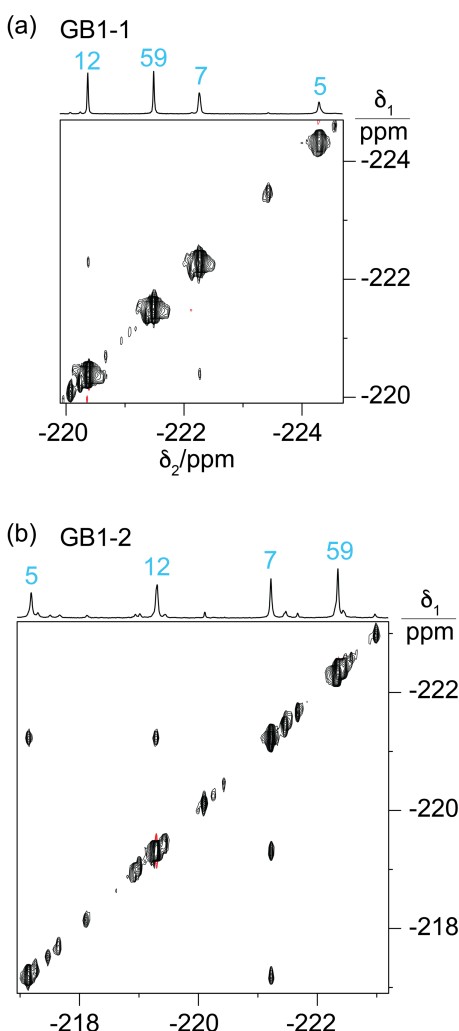

**Figure 7.** [$^{19}$F,$^{19}$F]-TOCSY spectra of 2 mM solutions of GB1-1 and GB1-2 recorded with 60 ms mixing time. The 1D NMR spectra are plotted on top with the resonance assignments in blue. (a) TOCSY spectrum of GB1-1 recorded in about 3 hours, using $t_{1max}$ = 8.5 ms and $t_{2max}$ = 128 ms. The cross-peaks between residues 7 and 12 are about 100 times weaker than the diagonal peaks. (b) TOCSY spectrum of GB1-2 recorded in about 12 h, using $t_{1max}$ = 8 ms and $t_{2max}$ = 171 ms. The cross-peaks are about 10 times smaller than the diagonal peaks.

## 3.5 Heteronuclear NMR for residue assignment

Heteronuclear [$^{1}$H,$^{19}$F]-NOESY (HOESY) spectra recorded with 150 ms mixing times showed NOEs with nearby protons (Fig. 8). These NOEs delivered residue-specific resonance assignments, as many of the corresponding $^{1}$H nuclei were also



detected in conventional homonuclear [$^1$H,$^1$H]-NOESY spectra. For example, residue 5 in GB1-1 and GB1-d displays NOEs to a $^1$H resonance at about -0.8 ppm. This resonance matches a b-proton of Leu5, which in wild-type GB1 is the most high-field $^1$H resonance due to aromatic ring currents from Phe28. In all three samples, the $^{19}$F-NMR signal of residue 5 produced

stronger HOESY cross-peaks than the other fluorinated leucine residues, while residue 59 delivered relatively weak cross-peaks if any. This result indicates that a CH$_2$F group produces stronger HOESY cross-peaks when it is buried in the core of the protein than when it is solvent exposed and can rotate in an unhindered manner. The $^{19}$F-NMR assignments of residue 7 were confirmed similarly by comparison of the cross-peaks observed in the HOESY and [$^1$H,$^1$H]-NOESY spectra.

In the case of the diFLeu residue in position 12, stronger cross-peaks were detected for the low-field signal assigned to the

C$^{\delta 2}$H$_2$F than the C$^{\delta 1}$H$_2$F group (Fig. 8c). In addition, the C$^{\delta 1}$H$_2$F group of this residue displays a negative cross-peak with the water resonance (at 4.75 ppm) as do both $^{19}$F-NMR signals of residue 59, indicating intermolecular NOEs with hydration water molecules featuring sub-nanosecond residence times (Otting et al., 1991). This confirms the solvent exposure of these fluorine atoms and agrees with the stereospecific assignments of residue 12 made by comparing the $^{TS}J_{FF}$ couplings with the protein structure.

Starting from the assignment of residue 5, the cross-peaks observed in the [$^{19}$F,$^{19}$F]-TOCSY spectra provided an additional, straightforward assignment pathway for the $^{19}$F spins in GB1-d and GB1-2 (Fig. 6b and 7b). In GB1-dd as in GB1-1, the C$^{\delta 1}$H$_2$F group of residue 12 produced only weak HOESY cross peaks. The HOESY spectrum thus did not identify the $^{19}$F-NMR signals belonging to the same diFLeu residue in position 12. This link, however, was easily established by correlations with the γ-proton of the isopropyl group observed in a short-delay $^1$H,$^{19}$F correlation experiment.




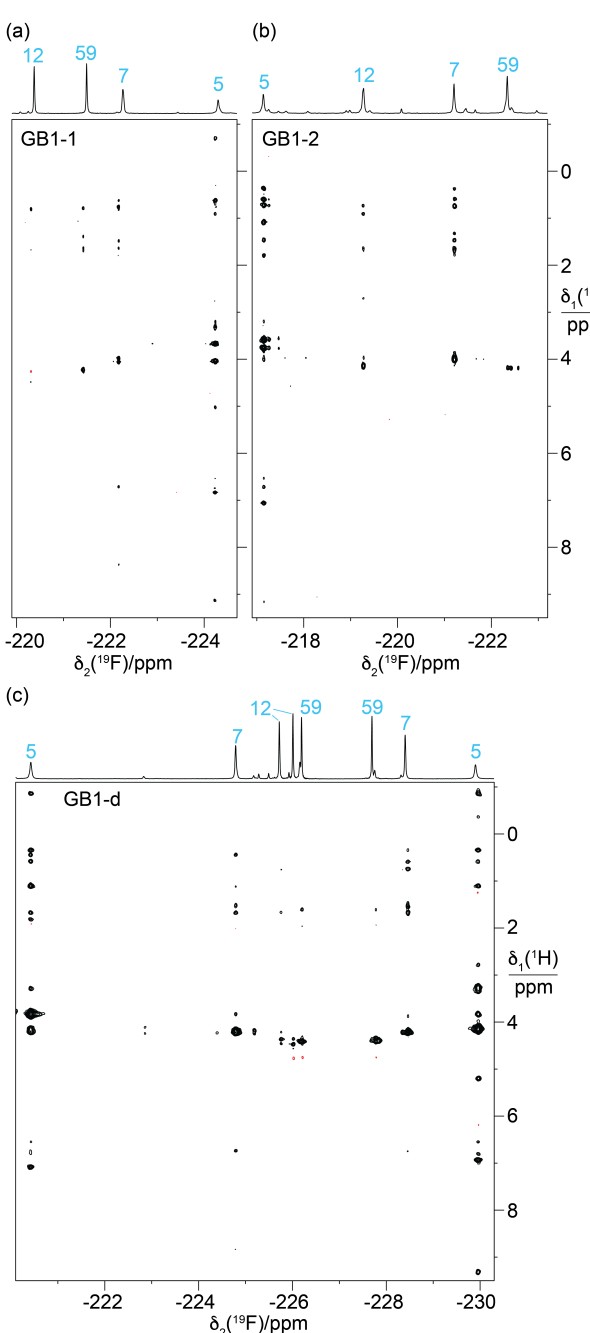

**Figure 8.** [$^1$H,$^{19}$F]-HOESY spectra of GB1 produced with FLeu1, FLeu2 or diFLeu. The spectra were recorded with a mixing time of 150 ms. The corresponding 1D $^{19}$F-NMR spectra are shown at the top along with the resonance assignments. (a) HOESY spectrum of a 2.2 mM solution of GB1-1, recorded using $t_{1max}$ = 38 ms, $t_{2max}$ = 136 ms, total recording time 9.6 h. (b)





HOESY spectrum of a 2 mM solution of GB1-2, recorded using $t_{1max}$ = 30 ms, $t_{2max}$ = 136 ms, total recording time 34 h. (c) HOESY spectrum of a 10 mM solution of GB1-d, recorded using $t_{1max}$ = 31 ms, $t_{2max}$ = 108 ms, total recording time 2.3 h.

### 3.6 Measurement of $^3J_{HF}$ couplings and γ-gauche effect

$^3J_{HF}$ couplings are governed by a Karplus relationship describing their dihedral angle dependence and thus provide information about the rotameric states of the $CH_2F$ groups. Quantitative measurements of $J$-couplings in the 1D $^{19}$F-NMR spectra recorded
without $^1$H decoupling were hampered by spectral overlap and the presence of sample heterogeneities (Fig. 3). Narrower $^{19}$F multiplets were obtained for samples prepared with diFLeu versions that had been synthesized with $CD_2F$ instead of $CH_2F$ groups (Maleckis et al., 2021), where $^3J_{HF}$ couplings were manifested in a sample made with diFLeu containing a $C^\gamma H$ group, whereas these splittings were absent from a sample prepared with deuterated diFLeu containing a $C^\gamma D$ group. A simple comparison of these spectra shows a correlation between the $^{19}$F chemical shifts and splittings due to $^3J_{HF}$ couplings (Fig. S4).
For more quantitative measurements of the $^3J_{HF}$ couplings, we recorded short-delay $^1$H,$^{19}$F correlation experiments (Tan et al., 2024), which encode the $^3J_{HF}$ coupling constants in the relative peak intensities of $H^\gamma$–$^{19}$F versus $H^\delta$–$^{19}$F cross-peaks. The results confirm the correlation between the $^3J_{HF}$ coupling constants and the $^{19}$F chemical shifts (Fig. 9). This correlation is a hallmark of the γ-gauche effect, which associates a high-field $^{19}$F chemical shift with the rotameric state of the $CH_2F$ group that positions the $^{19}$F spin *trans* relative to the γ-proton of the isopropyl group (Feeney et al., 1996; Tan et al., 2024; Frkic et
al., 2024a,b). Conversely, the $^{19}$F-NMR resonance is shifted low-field, if the $^{19}$F spin is positioned *trans* relative to a carbon atom. The γ-gauche effect is most clearly illustrated by the low-field and high-field signals of residue 5.

The immediate chemical environment of the $^{19}$F spins also affects their chemical shifts. For example, the $^3J_{HF}$ coupling of the high-field $^{19}$F resonance of residue 7 in GB1-dd is smaller than for residue 5, yet the resonance appears more high-field in the spectrum (Fig. 9d). Quite generally, the $^{19}$F chemical shifts of residue 7 are very sensitive to the presence or absence of
fluorinated residues in positions 5 and 12 (Fig. 9c and d), highlighting the impact of the chemical environment.

Interestingly, the $^{19}$F spins of residue 7 showed significantly different $^3J_{HF}$ couplings between the GB1-d and GB1-dd preparations, suggesting somewhat different populations of the different rotameric states of the $CH_2F$ groups. The associated changes in $^{19}$F chemical shifts are high-field and low-field as predicted by the γ-gauche effect.



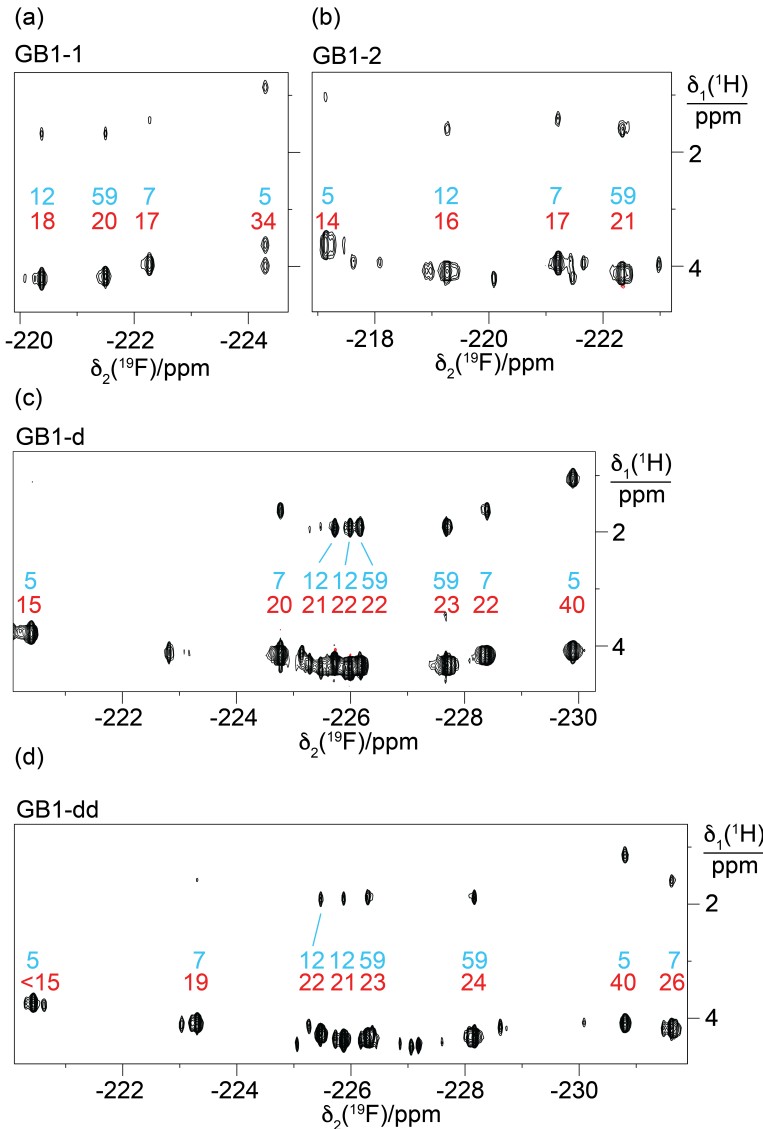

**Figure 9.** Short-delay $^1$H,$^{19}$F correlation experiments for the measurement of $^3J_{HF}$ coupling constants. The experiments were conducted with a $^1$H constant-time evolution period of 7 ms to evolve the $J_{HF}$ couplings and a refocusing INEPT period of 2.5 ms (Tan2024). The resonance assignments are indicated in blue. Red numbers indicate the $^3J_{HF}$ coupling constants (in Hz) derived from the relative intensities of the H$^\gamma$–$^{19}$F versus H$^\delta$–$^{19}$F cross-peaks. (a) Spectrum recorded of a 2 mM solution of GB1-1. Parameters: $t_{1max}$ = 7 ms, $t_{2max}$ = 128 ms, total recording time 1.3 h. (b) Spectrum recorded of a 2 mM solution of GB1-2, using $t_{1max}$ = 6.9 ms, $t_{2max}$ = 171 ms, total recording time 2.8 h. (c) Spectrum recorded of a 0.8 mM solution of GB1-d, using $t_{1max}$ = 7 ms, $t_{2max}$ = 180 ms, total recording time 5.3 h. (d) Spectrum recorded of a 0.6 mM solution of GB1-dd produced with diFLeu diluted with Leu, using $t_{1max}$ = 7 ms, $t_{2max}$ = 181 ms, total recording time 22.3 h.





Very large and very small $^3J_{HF}$ couplings indicate that $CH_2F$ groups are trapped in pure *trans* or *gauche* rotamers, respectively, showing that the rotation of a $CH_2F$ group about the $C^\gamma–C^\delta$ bond axis can be halted by the steric restraints in the tightly packed core of the protein. In the case of the *E. coli* peptidyl−prolyl *cis−trans* isomerase B (PpiB) produced with FLeu and diFLeu, we determined $^3J_{HF}$ couplings ranging between 9 and 44 Hz (Tan2024). The $^3J_{HF}$ couplings observed in GB1 are less extreme, suggesting that each $CH_2F$ group populates more than a single rotamer. Using residue 59 located in the flexible TEV cleavage

motif as a reference, a $^3J_{HF}$ coupling of about 22 Hz is indicative of a $CH_2F$ group that populates all three possible staggered rotamers. The different $^3J_{HF}$ couplings observed for residue 5, which is the most deeply buried leucine side chain in the wild-type protein, thus show clear conformational preferences for its $CH_2F$ groups. The largest $^3J_{HF}$ couplings were observed for diFLeu rather than FLeu1 or FLeu2 residues as expected for fluorine–fluorine interactions biasing the conformational space of the $CH_2F$ groups (Marstokk1997, Wu1998, Lu2019).

On a technical note, the short-delay $^1H,^{19}F$ correlation experiments delivered the $H^\gamma$ chemical shifts with much greater sensitivity than the [$^1H,^{19}F$]-COSY experiment recorded without heteronuclear decoupling (Fig. 5), assisting with the resonance assignments by comparison with [$^1H,^1H$]-NOESY spectra. In terms of sensitivity, the short-delay $^1H,^{19}F$ correlation experiments were also far superior to the HOESY spectra. The chemical shifts of the $H^\gamma$ spins were well conserved between the samples made with FLeu1, FLeu2 and diFLeu, ascertaining the $^{19}F$ resonance assignment of GB1-dd by comparison with

GB1-d.

### 3.7 $^{13}C$-NMR spectroscopy

The $^{13}C$ chemical shifts of the $CH_3$ groups in the singly fluorinated leucine analogues FLeu1 and FLeu2 were shifted upfield by between 5.6 and 8.2 ppm relative to the shifts of the methyl groups in the wild-type protein (Fig. S5). Highly conserved $^1H$ and $^{13}C$ chemical shifts of the GB1 variants indicate that the three-dimensional fold of the protein remains unchanged by the

fluorinated leucine analogues. Therefore, any differences in chemical shifts reflect local rather than global effects. The $^{13}C$-HSQC spectra showed the cross-peaks of the $CH_2F$ groups in the $^{13}C$-dimension near 90 ppm for GB1-1 and GB1-2, and about 86 ppm for GB1-d (Fig. S6). In the $^1H$-dimension, most $CH_2F$ groups displayed two different chemical shifts for the diastereotopic $^1H$ spins which, except for residue 5 in GB1-1, were unresolved in the short-delay $^1H,^{19}F$ correlation experiments (Fig. 9). The intensities of the $^{13}C$-HSQC cross-peaks of the $CH_2F$ groups of residue 5 were rather weak (similar to those of

$CH_2$ groups of other buried amino acid residues), which correlates with the relatively broad $^{19}F$-NMR signals observed for this residue. The other $CH_2F$ groups showed more intense $^{13}C$-HSQC cross-peaks on par with solvent-exposed $CH_2$ groups. The methyl cross-peaks of Leu5 are relatively weak also in wild-type GB1 (Fig. S5; Goehlert et al., 2004).



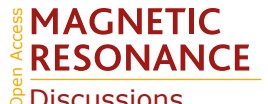

## 3 Discussion

The conformational impact of the fluorination of leucine methyl groups has previously been investigated in solution for only a single protein, PpiB, which contains five isolated leucine residues (Tan et al., 2024; Frkic et al., 2024a). The current findings recapitulate many of the findings made for PpiB.

(i) In the case of the most buried residue, residue 5, the rotation of the $CH_2F$ groups is sufficiently hindered to bias the populations of the different rotamers in favour of a *trans* configuration of the $C^{\delta 1}H_2F$ group and a *gauche* configuration of the

$C^{\delta 2}H_2F$ group as defined by the $^3J_{HF}$ coupling constants. The size of the $^3J_{HF}$ couplings indicates that these conformational biases are more pronounced in GB1-d than in GB1-1 and GB1-2, which may be attributed to unfavourable electrostatic interactions between parallel and antiparallel C–F bonds in a 1,3-difluoropropane moiety intrinsically limiting the conformational freedom (Marstokk and Møllendal, 1997; Wu et al., 1998; Lu et al., 2019).

(ii) The large chemical shift dispersion of the $^{19}F$ NMR signals over many ppm is mainly due to the γ-gauche effect, which

attributes high-field and low-field shifts to *trans* and *gauche* rotamers (Feeney et al., 1996). Intermediate chemical shifts correlate with intermediate $^3J_{HF}$ coupling constants and are thus indicative of averaging between different rotamers. To the best of our knowledge, the present work is only the third experimental example of the γ-gauche effect (Tan et al., 2024; Frkic et al., 2024b).

(iii) The more solvent accessible $CH_2F$ groups displayed less extreme $^3J_{HF}$ couplings and less extreme $^{19}F$ chemical shifts,

suggesting more extensive averaging between different rotameric states. Larger and smaller $^3J_{HF}$ couplings as well as greater $^{19}F$ chemical shift dispersions have been observed previously in PpiB (Tan et al., 2024), suggesting that the $CH_2F$ groups populate more than a single rotamer even in the buried residue 5.

(iv) The line widths of the $^{19}F$ NMR signals vary greatly between different residues and, most strikingly for residue 12, between different $CH_2F$ groups. Narrow signals correlate with high solvent exposure. For wild-type GB1, order parameters $S^2_{axis}$

determined by relaxation measurements have been reported for the methyl groups of Leu12 (< 0.15), Leu7 $C^{\delta 2}H_3$ (0.15) and Leu5 $C^{\delta 1}H_3$ (0.55), showing that the methyl group symmetry axes are subject to motions, which are more prominent in situations of high solvent exposure (Goehlert et al., 2004). In agreement with this result, the $^{19}F$-NMR signals of Leu5 are broader than any others.

(v) For any given position in the protein, the relative chemical shifts observed between FLeu1 and FLeu2 are strongly

predictive of the stereospecific assignments of a diFLeu residue at the same site. The same feature also prevails in PpiB (Tan et al., 2024).

The present work shows, for the first time, that through-space $^{19}F$–$^{19}F$ couplings can readily be detected between singly fluorinated $CH_2F$ groups in a protein. In previous work, we detected $^{TS}J_{FF}$ couplings between genetically encoded $CF_3$-phenylalanine and $CF_3$-tyrosine residues in the core of PpiB (Orton et al., 2021). Notably, however, the $^{19}F$ NMR spectra of

PpiB constructs with multiple $CF_3$ groups showed additional resonances suggesting structural perturbations arising from the additional space requirements of $CF_3$ groups. The $^{19}F$-NMR spectra of GB1 made with fluorinated leucine analogues also





display weak additional resonances, but there is no evidence that they are due to structural heterogeneity. Instead, the additional signals are consistent with chemical heterogeneity arising from incomplete substitution of canonical leucine by fluorinated leucines or incomplete optical purity of the synthesized fluoroleucine. Cell-free protein synthesis enables the requisite high

level of global substitution of canonical amino acids by fluorinated analogues.

The observation of $^{TS}J_{FF}$ couplings in GB1 is non-trivial as they depend on direct contact between the fluorine atoms. Crystal structures of PpiB showed that $CH_2F$ groups often populate multiple staggered rotamers that differ by rotation about the bond with the carbon atom they are bound to (Frkic et al., 2024a,b). Based on the 3D structure of wild-type GB1 (Fig. 2), only specific rotamer combinations generate fluorine–fluorine contacts. A crystal structure of ubiquitin synthesized chemically with

two FLeu1 residues indicated that the lowest energy conformation avoids fluorine–fluorine contacts (Alexeev et al., 2003). In the case of 1,3-difluoropropane, it is known that the polarity of C–F bonds discourage rotamers that produce fluorine–fluorine contacts (Marstokk and Møllendal, 1997; Wu et al., 1998; Lu et al., 2019). Therefore, the privileged attraction between fluorine atoms in perfluorinated polymers such as Teflon does not govern the interaction between the single fluorine atoms of $CH_2F$ groups. Nonetheless, the transient $^{19}F$–$^{19}F$ contacts arising from random rotations of the $CH_2F$ groups in GB1-1, GB1-2 and

GB1-d suffice to generate observable $^{TS}J_{FF}$ couplings. As noted previously (Orton et al., 2021; Tan et al., 2024), the much steeper distance dependence of $^{TS}J_{FF}$ couplings compared with $^{19}F$–$^{19}F$ NOEs strongly favours the detection of transient fluorine–fluorine contacts by [$^{19}F$,$^{19}F$]-TOCSY rather than [$^{19}F$,$^{19}F$]-NOESY experiments (Fig. 6).

For the side chain of Leu12 in GB1, a very different $c_2$ angle has been reported by the crystal structures (1PGA, 1PGB; Gallagher et al., 1994; 2QMT; Frericks Schmidt et al., 2007) versus the NMR solution structure (3GB1; Juszewski et al.,

1999). As a result, the crystal structures expose the $d_2$ methyl group to the solvent, while the solution structure exposes the $d_1$ methyl group. The observation of $^1H$–$^{19}F$ NOEs with water together with different $^{19}F$-NMR line widths indicative of more facile rotation of the $C^{\delta1}H_2F$ than $C^{\delta2}H_2F$ group fully agree with the conformation of Leu12 depicted in Fig. 2, indicating that fluorinated leucine residues do not alter the solution structure. Simple rotations of the $CH_2F$ groups allow accommodating the fluorine atoms in the energetically most favourable rotamers.

Establishing sequence-specific resonance assignments of the $^{19}F$-NMR spectra by 2D NMR techniques rather than site-directed mutagenesis is straightforward for small proteins like GB1. For larger proteins, site-specific selective installation of the fluorinated amino acids by genetic encoding (Orton et al., 2021; Qianzhu et al., 2020; 2022; 2024) will be helpful. Work towards this goal is in progress.

**Data availability.** The NMR data are available at https://doi.org/10.5281/zenodo.14984603.



**Author contributions.** YJT, EHA and IDH prepared the protein samples and performed 1D NMR measurements. AM
synthesised fluorinated leucine analogues with and without deuteration. GO coordinated the project, performed the 2D NMR
measurements and prepared the final manuscript and figures.

**Acknowledgements.** We thank Dr Eliza Tarcoveanu for initial syntheses of FLeu1 and FLeu2.

**Financial Support.** This research has been supported by the Australian Research Council (grant no. DP230100079) and the
Australian Research Council Centre of Excellence for Innovations in Peptide and Protein Science (grant no. CE200100012).

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
