# Peer review of "Inter-residue through-space scalar 19F–19F couplings between CH2F groups in a protein"

_Magnetic Resonance, 2025_

## Author Comment (AC1)

We recorded a [$^{19}$F,$^{19}$F]-TOCSY spectrum of GB1 made with deuterated 5,5'-difluoro-L-leucine (where all protons of the fluorinated isopropyl groups are replaced by deuterons). The spectrum, shown underneath, shows all the strong cross-peaks of the spectrum of Figure 6a in the main text.

[Figure]

Figure 1. [$^{19}$F,$^{19}$F]-TOCSY spectrum of a 0.5 mM solution of GB1 made with deuterated 5,5'-difluoro-L-leucine. Parameters: mixing time 55 ms, $t_{1max}$ = 6 ms, $t_{2max}$ = 97 ms, total recording time 15 h, 400 MHz NMR spectrometer.

The article by Blake et al. (J. Biomol. NMR, 2, 527, https://doi.org/10.1007/BF02192814, 1992) reported small through-space scalar couplings in rubredoxin between the methyl protons of Ala 43 and heavy metal ions ($^{113}$Cd, $^{199}$Hg). In our case, we find no sign of through-space couplings between $^{19}$F and other spins. If they were a common occurrence, we would expect to observe through-space $^{19}$F-$^{1}$H couplings for some of the protons that show the strongest NOEs in the HOESY spectra. Neither the COSY spectrum of Figure 5 nor the short-delay $^{1}$H,$^{19}$F correlation experiments showed any evidence for this. Fluorine bonded to carbon features more hydrophobic than H-bond acceptor characteristics. The leucine methyls of GB1 are not near H-bond donors or heavy atoms.

To the best of our knowledge, scalar $^{19}$F-$^{19}$F couplings mediated by electrons of an intervening chemical group have been reported only once, where the intervening group was the $\pi$-bond system of a phenyl ring, see Mallory et al. (J. Am. Chem. Soc., 112, 2577, https://doi.org/10.1021/ja00163a015, 1990).

---

## Author Comment (AC2)

**Table.** $R_{1\rho}(^{19}F)$ values measured in CPMG experiments with different spacing between 180° pulses and comparison with R2* values.[a]

| Sequence position and stereospecific assignment | GB1-d, $R_{1\rho}$ with 1250 Hz CPMG[b] | GB1-d, $R_{1\rho}$ with 31 Hz CPMG[c] | GB1-d-D, $R_{1\rho}$ with 1250 Hz CPMG[d] | GB1-d, $R_2$* [e] |
|---|---|---|---|---|
| 5, δ2 | 31 (1) | 37 (3) | 15 (1) | 53 |
| 7, δ2 | 15.9 (0.2) | 25 (1) | 10 (1) | 24 |
| 12, δ2 | 8.3 (0.5) | 11.7 (0.2) | 3.1 (0.5) | 23 |
| 12, δ1 | 6.8 (0.4) | 8.6 (0.1) | 3.7 (0.5) | 23 |
| 59, δ1 | 7.3 (0.2) | 12.5 (0.7) | 4.9 (0.3) | 21 |
| 59, δ2 | 7.8 (0.7) | 11.2 (0.4) | 3.6 (0.4) | 21 |
| 7, δ1 | 13.6 (0.2) | 26 (1) | 9.1 (0.4) | 29 |
| 5, δ1 | 37 (1) | 33 (4) | 20 (1) | 56 |

[a] Measured on a 400 MHz NMR spectrometer. $R_{1\rho}$ and $R_2$* values given in s$^{-1}$, showing the standard deviation of fitting in brackets. $R_2$* values were determined by measuring the full width at half height in 1D $^{19}$F-NMR spectra (in Hz) and multiplying by $\pi$. GB1-d refers to the samples made with diFLeu residues. GB1-d-D refers to samples made with diFLeu residues, where all five protons of the isopropyl group are replaced by deuterium.

[b] Using 0.4 ms spacing between 180° pulses. Relaxation delays used were 0.8, 16.8, 32.8, 40.8, 56.8, 72.8, 88.8, 104.8 ms.

[c] Using 16 ms spacing between 180° pulses. Relaxation delays used were 32, 48, 64, 80, 96, 112, 128, 144 ms.

[d] Using 0.4 ms spacing between 180° pulses. Relaxation delays used were 0.8, 7.2, 13.6, 20, 26.4, 32.8 39.2, 45.6, 52, 58.4, 64.8, 71.2, 77.6 ms.

[e] 1D $^{19}$F-NMR spectrum measured with 2800 Hz GARP decoupling of $^1$H during 144 ms data acquisition.

[Figure]

**Figure.** Plots of the function $I_C/I_D = \tan^2(\pi J_{FF}\tau_m)$ versus the TOCSY mixing time $\tau_m$ for three cross-peaks between $CD_2F$ groups in GB1-d made with the diFLeu version, where all five protons of the isopropyl group are replaced by deuterium (Figure S4). $I_C$ and $I_D$ denote the integrals of, respectively, the cross-peaks and diagonal peaks measured in a one-dimensional cross-section. The cross-peaks are between residues 5 and 7 (a and b), and within the solvent-exposed residue 59 (c). Subscripts report the stereospecific assignments of the $CD_2F$ groups. The $J_{FF}$ couplings determined from best fits are indicated. The fits were based on the simplifying approximation of a 2-spin system, where the cross-peak intensity grows with $\sin^2(\pi J_{FF}\tau_m)$ and the diagonal peak decays with $\cos^2(\pi J_{FF}\tau_m)$ (Braunschweiler and Ernst, 1983).

**Reference**

Braunschweiler, L. and Ernst, R. R.: Coherence transfer by isotropic mixing – application to proton correlation spectroscopy, J. Magn. Reson. 53, 512–528, https://doi.org/10.1016/0022-2364(83)90226-3, 1983.

---

## Author Comment (AC3)

**Figure.** 1D $^{19}$F-NMR spectra of two different preparations of GB1-d. (a) 3 mM solution in 50 mM HEPES buffer, pH 7.5, 100 mM NaCl. (b) 1 mM solution in 20 mM MES buffer, pH 6.5, 100 mM NaCl. The $^{19}$F chemical shifts are insensitive to the solution conditions.

---

## Author Response (AR1)

In response to the reviewers' comments, we have made the changes detailed below. For some of the points, more detailed responses have been given during the discussion phase of the preprint.

In response to Ad Bax's comments:

1. I'm a bit confused about the gamma gauche effect when talking about two monofluorinated methyl groups in Leu. Both 19F nuclei can simultaneously be trans relative to Cgamma, so it's not clear that this helps with stereo assignment, even though the correlation between JHF and 19F shift clearly shows it to be correct. Is it conceivable that Calpha_Cdelta gamma gauche effects contribute to the 19F chemical shift (as they do for 13Cdelta)?

Response: Obviously, the preservation of high-field and low-field shifts between singly and doubly fluorinated leucines is a purely empirical observation prone to exceptions. Nonetheless, it is unlikely that $CH_2F$ groups populate the same rotamers in monofluoro- and difluoro-leucine residues, as the conformations with two parallel C-F bonds are energetically disadvantaged (Marstokk and Møllendal, 1997).

Fluorination seems to override the $C\alpha$-$C\delta$ $\gamma$-gauche effect for the $CH_2F$ groups in GB1-1 and GB1-2: Figure S7 shows that the $^{13}C$ chemical shift of the $C^{\delta 1}H_2F$ group of residue 12 is greater than that of the $C^{\delta 2}H_2F$ group, although (according to the solution structure 3GB1) the latter carbon is trans to the $\alpha$-carbon in the wild-type protein. The $C\alpha$-$C\delta$ $\gamma$-gauche effect and the reference https://doi.org/10.1007/BF00202043 (MacKenzie et al., 1996) are now cited in line 365.

2. For the HOESY measurements, I suspect the NOE effect to be very sensitive to internal motion due to the closeness of wH and wF. Heteronuclear 1H-19F NOE can be negative or positive, depending on applicable spectral densities and some comments may be helpful.

Response: Internal motions of $CH_2F$ groups are more restricted than for methyl groups. In 1-fluoropropane the intrinsic energy barrier between staggered rotamers has been calculated to be 4-5 kcal/mol (Feeney et al., 1996), which is comparable to the energy barrier for rotation about the central C-C bond in n-butane. Therefore, $^1H$-$^1H$ NOEs with $CH_2F$ groups are not intrinsically different from other $^1H$-$^1H$ NOEs. Experimentally, we did not observe negative $^1H$-$^{19}F$ cross-peaks with the $CH_2F$ groups apart from NOEs with water, which we now point out in a sentence on line 267.

3. 19F line widths are reported to be 7-15 Hz, which corresponds to R2 values of 20-40/s, but this seems fast considering the 60-ms TOCSY mixing times used. Could incomplete decoupling or isotope effects contribute to these line widths? Perhaps a R1rho number would be helpful.

Response: After double-checking, we corrected the upper limit to 18 Hz. We now report $R_{1\rho}(^{19}F)$ values of GB1-d and GB1-d-D (which is the same as GB1-d except for

having been made with deuterated diFLeu) in the new Table S3. The slower $R_{1\rho}(^{19}F)$ relaxation rates observed for GB1-d-D suggest that dipolar relaxation by the nearest protons contributes significantly to the $^{19}F$ relaxation. This is now reported in two sentences on lines 132 to 136.

As the solvent contained 90% $H_2O$/10% $D_2O$, deuterated amides could indeed contribute to isotope effects. Measurements in the presence of less $D_2O$ (2%), however, did not result in narrower signals.

4. Line 146-147: "faster rotation of the CH2F group about the Cgamma-Cdelta bond results in slower transverse relaxation".  This may well be true, but the magnitude of this effect seems larger than expected considering the modest chemical shift difference of ~10ppm.  Could crank-shaft sidechain motions, previously suggested to be responsible for different Cdelta 13C relaxation rates, play a role?

Response: We find it hard to picture a crankshaft motion about the $C\gamma$-$C\delta$ bond that would be equally benign in terms of structural conservation as a simple rotation of the $CH_2F$ group, at least for motions changing the conformation to a similar degree. We imagine that the slopes of the potential wells between different staggered rotamers are steeper for $CH_2F$ groups than $CH_3$ groups because of the greater energy barrier between them. Our sentence (now line 171) *suggests*, rather than claims, that the line width is governed by the speed of rotation of the $CH_2F$ groups.

5. Line 199: "different" from what?  Perhaps use "multiple"?

Response: We now use the term "multiple" (now line 220).

6. Line 201: "greater conformational freedom than suggested by 3GB1". This is a bit of a philosophical issue, but the width of an "NMR bundle" does not reflect motional freedom but the certainty at which the structure that agrees best with NMR restraints can be determined.  If not, measuring fewer restraints would make the protein more dynamic.

Response: We now point out on line 223 that the NMR structure was determined with the aim of presenting the single best approximation to the average structure.

7. Line 322: "the third example of the gamma-gauche effect". Probably correct, but perhaps useful to remind the reader of how commonly this is used in 13C analysis, including proteins (e.g. https://link.springer.com/article/10.1007/BF00202043 )

Response: We meant the $\gamma$-gauche effect only with regard to $CH_2F$ groups (now clarified on line 366). We also cite the reference of MacKenzie et al. (1996) on line 365.

8.  There is a considerable amount of older literature on TS-JFF couplings, with an empirically determined very steep distance dependence. See e.g. Bakhmutov and references therein. Perhaps including some reference to this historic work would be helpful, e.g. https://doi.org/10.1002/mrc.1260231117

Response: We now refer the reader to the comprehensive review by Hierso (2014) on line 35 and cite a 1978 article for a large $^{TS}J_{FF}$ coupling.

9.  Lines 358-362, c=chi; d=delta

Response: Typos now fixed.

10. Please include the RF field strength and mixing scheme (DIPSI?) used for the TOCSY spectrum.

Response: This information is now provided in line 205.

11. Can the authors provide approximate TS-JHH values based on the cross/diagonal peak ratios?

Response: We measured the ratio of cross-peaks versus diagonal peaks in FF-TOCSY experiments recorded with increasing mixing times. The new Figure S5 shows curves fitted to three prominent cross-peaks, which indicate that the through-space $J_{FF}$ couplings in GB1-d are up to 2-3 Hz. The results are described in the new section 3.5.

12. Trivialities: Spell out CFPS upon first use; Juszewski is really Kuszewski (3 times).

Response: We corrected the spelling.

In response to reviewer 2:

1.  The authors observe 'through-space' J-couplings between $^{19}F^{\delta}$ atoms of the leucines, which are supposedly in spatial contact. Due to the Fermi contact mechanism, the J-couplings show that the respective s-electrons are correlated in their motion. How do the authors imagine this electronic polarization transfer? Given that the fluorines have strong negative partial charges, would they interact directly in an attractive or repulsive way? Could it also be that the hydrogens in the $CFH_2$ group acquire partial positive charges and are then attracted to the fluorine of the neighboring leucine, such that the transfer is achieved as F...H-C-F? Maybe this is unlikely, since Teflon is very hydrophobic. An educated discussion would be very helpful.

Response: The quantum mechanical underpinning of through-space $^{19}$F-$^{19}$F $J$-couplings is complicated. We now refer to the 2013 review by Hierso in Chem. Rev.

Small through-space couplings between $^{19}$F and $^{1}$H would be difficult to measure given the relaxation rates of CH$_2$F groups. We would expect a F..H-C-F interaction between $^{19}$F spins to be weaker than a direct F-F interaction. Furthermore, C-F groups are known not to be good H-bond acceptors and the protons of CH$_2$F groups are barely acidic. Instead of speculating we cite the finding by Alexeev et al. (2003) that fluorine atoms of CH$_2$F groups are not specifically attracted to each other (line 395).

2. Can the authors estimate the size of the J$_{FF}$-couplings from the intensities of the TOCSY-spectra?

Response: We have done this. The results are reported in the new section 3.5.

3. The spectra in Figure 3 were recorded at different pH and buffer for the different GB1 analogues (pH 6.5 MES vs pH7.5 HEPES). Is there a reason? How much would the pH affect the chemical shifts? This should be discussed in the text. Also, the Figures of other spectra do not mention the pH and buffer conditions. They should be indicated throughout.

Response: It was an accident that different buffers were used for different preparations. Fortunately, the $^{19}$F-NMR spectra did not vary significantly with the buffer (now stated in line 112; see also the spectra in the supplement to our response to reviewer 2 in the discussion phase). On double-checking, the samples made with diFLeu consistently were in MES buffer, those with FLeu2 consistently were in HEPES buffer, those with FLeu1 mostly in HEPES buffer unless reported otherwise. This is now clarified in section 2.6. As GB1 contains no histidine in the structured part of the protein, also the $^{1}$H chemical shifts were conserved between the pH values of the different buffers (6.5 and 7.5).

4. The authors mention $^{19}$F T$_1$ relaxation times of 0.3 s (line 121). Can they estimate the T$_2$s? They indicate line widths of 7–15 Hz. Would this be 20–50 ms? Do they have information from spin-echos?

Response: Transverse relaxation rates are now reported in Table S3.

5. Figure 9: a plot of the $^3$J$_{HF}$ constants vs. the $^{19}$F shifts would be very helpful to visualize and quantify the γ-gauche effect.

Response: The plot is now shown in Figure S8.

6. Apparently the GB1 construct contains an additional MASMGT sequence at the N-terminus. Although no effect is expected, it would be good to mention this in the main text.

Response: This is now spelled out in line 62.

7. Figure S3: CD melting. Is there a reason for the larger (absolute) ellipticity of wild-type GB1? Please mention/discuss.

Response: We attribute the differences to different sample concentrations. We now state in the legend of Figure S3 that the sample concentrations were *about* 0.3 mg/mL. The $^1$H NMR data leave no doubt about the structural integrity of the fluorinated samples.

8. Figure S5: is this a natural-abundance $^{13}$C HSQC? This should be mentioned in the main text.

Response: Yes, natural abundance, which is now stated in the legends of Figures S5 and S6.

Presentation:

The accessibility of the manuscript could be improved by considering the following:

1. The nomenclature is hard to follow. (2S,4S)-5-fluoroleucine, (2S,4R)-5-fluoroleucine and 5,5'-difluoro-L-leucine are FLeu1, FLeu2 and diFLeu. Apparently FLeu1, FLeu2 and diFLeu are leucines labeled in the $\delta$1, $\delta$2 or $\delta$1+$\delta$2 respectively (if I am not mistaken). This should be clearly indicated in the drawing of Figure 1 (not in the legend). A standard IUPAC NMR nomenclature of all leucine atoms would be also very helpful in this Figure. Could FLeu1, FLeu2 and diFLeu be replaced by something like $F^{\delta 1}$Leu, $F^{\delta 2}$Leu, $F^{\delta 1 \delta 2}$Leu throughout the text?

Response: We changed Figure 1 to show the $\delta$1 and $\delta$2 labels. In general, we prefer not to relegate the stereospecificity information to a small superscript.

2. Throughout the figures: the authors are very terse with labeling in the figure graphics. While this emphasizes the data, it makes them hard to grasp. In particular:

2.1. Figure 2: indicating the $\delta$1, $\delta$2 color code directly in the graphic would be very helpful.

2.1. All 1D spectra: the stereospecific assignments would be better indicated by $\delta$1, $\delta$2 instead of a red dot only for $\delta$1.

2.3. All 2D spectra: individual peaks should be labeled with 2D, stereospecific assignment information whenever possible.

2.5. Figure S5: please provide color code in graphic.

Response: We now provide a colour code in Figure 2.

In the 1D NMR spectra, we used red dots instead of $\delta 1$ labels to make it graphically obvious that the $^{19}$F chemical shift of the $\delta 1$ group can be both high-field and low-field of the $\delta 2$ signal. In addition, some of the red dots come close to each other, leaving insufficient space for labelling with characters.

We believe that labelling every cross-peak in the 2D spectra would lead to unhelpful clutter.

The colour code has been added in Figure S5.

1. line 286: please indicate the size of the 'different 3JHF for residue 5'.

Response: We now quote the couplings of the most unexpected outlier and refer to Table S4 (line 298).

2. lines 358–360: the Greek characters were lost.

Response: We fixed the typos.